# Thermal, Mechanical, and Morphological Characterisations of Graphene Nanoplatelet/Graphene Oxide/High-Hard-Segment Polyurethane Nanocomposite: A Comparative Study

**DOI:** 10.3390/polym14194224

**Published:** 2022-10-09

**Authors:** Muayad Albozahid, Haneen Zuhair Naji, Zoalfokkar Kareem Alobad, Jacek K. Wychowaniec, Alberto Saiani

**Affiliations:** 1Department of Materials Engineering, Faculty of Engineering, University of Kufa, Najaf 54003, Iraq; 2Department of Chemical Engineering, Faculty of Engineering, University of Babylon, Hilla 51002, Iraq; 3Department of Polymers Engineering and Petrochemical Industries, Faculty of Materials Engineering, University of Babylon, Hilla 51002, Iraq; 4Department of Materials, Manchester Institute of Biotechnology, School of Natural Sciences, Faculty of Science and Engineering, The University of Manchester, Manchester M13 9PL, UK

**Keywords:** high hard-segment polyurethane, graphene nanoplatelets, graphene oxide, nanocomposites, thermal stability, mechanical properties

## Abstract

The current work investigates the effect of the addition of graphene nanoplatelets (GNPs) and graphene oxide (GO) to high hard-segment polyurethane (75% HS) on its thermal, morphological, and mechanical properties. Polyurethane (PU) and its nanocomposites were prepared with different ratios of GNP and GO (0.25, 0.5, and 0.75 wt.%). A thermal stability analysis demonstrated an enhancement in the thermal stability of PU with GNP and GO incorporated compared to pure PU. Differential Scanning Calorimetry (DSC) showed that both GNP and GO act as heterogeneous nucleation agents within a PU matrix, leading to an increase in the crystallinity of PU. The uniform dispersion and distribution of GNP and GO flakes in the PU matrix were confirmed by SEM and TEM. In terms of the mechanical properties of the PU nanocomposites, it was found that the interaction between PU and GO was better than that of GNP due to the functional groups on the GO’s surface. This leads to a significant increase in tensile strength for 0.5 wt.% GNP and GO compared with pure PU. This can be attributed to interfacial interaction between the GO and PU chains, resulting in an improvement in stress transferring from the matrix to the filler and vice versa. This work sheds light on the understanding of the interactions between graphene-based fillers and their influence on the mechanical properties of PU nanocomposites.

## 1. Introduction

It is critical to design high-performance polymer nanocomposites by achieving a uniform dispersion of nanofillers and strong interfacial adhesion between nanofillers and polymer matrices in nanocomposite systems [1,2,3]. Recently, many researchers have focussed their attention on the use of various types of nanofillers such as graphene, carbon nanotubes, and nanoclays, among others [4,5,6]. Nanocomposites’ performance depends on the substantial features of the nanofiller (the geometrical dimensions, surface area, aspect ratio, etc.) and the dispersion/distribution state of the nanofiller, as well as nanofiller–nanofiller and nanofiller–polymer interactions [5,7]. Poor dispersion and weak interfacial bonding still pose a challenge in the design of effective carbon nanofiller–polymer matrix nanocomposites [8].

Polyurethanes (PUs) are semicrystalline copolymers consisting of alternating hard and soft segments [9,10,11,12]. Most PUs are formed from polyether/polyester polyols along with aliphatic or aromatic di-isocyanates [6,13,14]. Indeed, a phase-separated microstructure is formed in PUs due to the thermodynamic incompatibility between hard segments and soft segments, providing a unique and outstanding mechanical performance [12]. A suitable choice of raw chemicals and dispersion methods leads to improved mechanical, thermal, and electrical properties [3,13]. Polyurethane (PU) has various applications, including fibres, coatings, adhesives, and smart actuators [7,15,16,17,18].

Graphene is a two-dimensional sheet with a two-dimensional honeycomb arrangement of carbon atoms [19]. Graphene and its derivatives are considered promising nanofillers for high-performance polymeric nanocomposite materials due to their unique mechanical, rheological, thermal, and electrical properties [2,20,21,22]. In fact, graphene usage has increasingly grown in different industrial applications over the past decade [23]. In particular, graphene nanoplatelets (GNPs) are often used as nanofillers, consisting of a few layers of graphene sheets [24]. Graphene oxide (GO) has also been considered one of the most promising multifunctional fillers derived from graphene [25]. The production of GO leads to oxygenated (hydroxyl, epoxide, and carboxyl groups) defects in its lattice [18,26,27], which affect the final properties of GO.

Interfacial interactions between nanofillers and polymer chains can provide a feasible way to build high-performance polymer nanocomposites [3,4,7]. Dispersion approaches can basically be classified into three methods: (i) melt blending, (ii) solution mixing, and (iii) in situ polymerisation [3,28]. The degree of dispersion between these methods can be varied and depends on the exact physicochemical interactions between the polymer and the nanofillers [29].

The intermolecular interactions during this process can lead to the restacking of 2D nanomaterials driven by the Van der Waals interactions when at close proximity [30]. For instance, high shear stress during melt mixing was reported to be inadequate for overcoming the agglomeration of nanofillers during mixing [31]. Consequently, in situ polymerisation can be a better choice for the high dispersion of nanofillers [26,32].

There are a significant number of studies devoted to the use of GNP, GO, or both to reinforce various kinds of polyurethane [26,33,34,35]. The reinforcements have shown improved electrical conductivity, mechanical properties, and thermal properties compared to pure PUs [36,37]. Graphene and GO are both advanced carbon nanomaterials that have a high mechanical strength and can be easily functionalised to tune interactions with polymers. Based on the authors’ knowledge, the dispersion and interaction of these nanofillers in polymer matrices could effectively improve crucial properties such as the mechanical and thermal properties of different polymers [38]. There is currently a lack of studies on this type of polyurethane with two types of nanoscale fillers with different functional groups on their surfaces. As such, the main objective of the current study is to obtain and characterise new hard-segment polyurethane nanocomposites with a new type of chain extender, 1,5 pentanediol, and using two kinds of nanoscale additives with different functional groups: GNP and GO.

## 2. Experimental Method

### 2.1. Materials

Graphene nanoplatelets (GNP), with an average thickness of 6–8 nm and a typical surface area of 120 to 150 m^2^/g, were purchased from Lansing, MI, USA (XG sciences.com). Polyurethane was produced according to a two-step polymerisation process using polyether polyol as a soft segment purchased from Sigma-Aldrich, UK, with functionality of 2.0 and an average molecular number M_n_ = 2000 g/mol. The chain extender (1,5-Pentanediol) and isocyanate, MDI (4,4’ methylenebis phenyl isocyanate), which represent the hard segments, were purchased from Sigma-Aldrich, UK. A catalyst (DABCO-S) and the solvent N,N dimethylene-acetamide (DMAC) were utilized to complete production of the PU.

### 2.2. Preparation of Graphene Nanoplatelets (GNP)

GNP powder (wt.%) was dispersed with DMAC solvent before mixing with PU at a concentration of 2 mg/mL. A sonication bath method (frequency = 37 KHz) was utilised for dispersion for half an hour to prevent GNP aggregation/stacking during the in situ polymerisation process.

### 2.3. Preparation of Graphene Oxide (GO)

Graphene oxide (GO) was prepared using a modified Hummers’ method [39]. Briefly, 5 g of natural flake graphite (Graphite Trading Company) and NaNO_3_ (3.75 g) were continuously stirred in concentrated sulphuric acid (170 mL). The mixture was then cooled and kept in an ice bath for 2 h with stirring to maintain a very low temperature of 2 °C. Afterwards, KMnO_4_ (25 g) was gradually added over 60 min with constant stirring and, to avoid a sudden increase in temperature, the rate of addition was carefully controlled. Then, the mixture was removed from the ice bath and allowed to warm to room temperature and left stirring for 7 days. The mixture became thicker and brownish over time. Then, the dark mixture was gradually dispersed into 550 mL of water with 15 mL (5 wt.%) of H_2_SO_4_ for approximately 1 h. Over 5 min, 10 mL of hydrogen peroxide (30 vol. %) was added with considerable effervescence; the mixture converted into a yellow/gold glittery suspension and was stirred for a further 2 h. After that, the mixture was additionally diluted with 500 mL of H_2_O containing 9 mL (3 wt.%) of H_2_SO_4_ and 2 mL (0.5 wt.%) of H_2_O_2,_ and left stirring overnight. Finally, the subsequent addition of water continued until the pH of the GO solution reached neutral (pH = 7).

### 2.4. Dispersion Techniques

PU nanocomposites were prepared using different dispersion methods in order to obtain better dispersion quality and an improvement in their physical properties [40]. The final product of synthesis was a solution-based PU with a high amount of DMAC solvent incorporated with GNP and GO by different dispersion techniques. Both GNP and GO were dispersed in DMAC using a sonication bath and magnetic stirring before being added to the PU during the synthesis process to ensure better quality dispersion. It was found that a sonication bath (power = 80 W; frequency = 37 kHz) of these carbon additives can exhibit a high degree of exfoliation compared to a magnetic stirrer (overnight). The difference between the sonicated and magnetic stirring-treated (GO and GNP) nanofillers is shown via the images in Figure 1.

### 2.5. Preparation of PU/GNP and PU/GO Nanocomposites

A two-step polymerisation approach was carried out to synthesise the PU matrix, as reported in our previous study [21], with a hard-segment ratio of 75% HS. The in situ polymerisation approach was used for the preparation of PU/GNP and PU/GO nanocomposites with a weight fraction value, for example, of 0.25 g of GNP/GO, added to 100 g of an exact amount of PU polymer for a ratio of 0.25 wt.%, and so on, since the GNP/GO solution was combined with the chain extender in the second stage. After completing the synthesis process, the PU/GNP and PU/GO solutions were dried in a furnace at 80 °C for three days. The test samples of both PU/GNP and PU/GO nanocomposite materials were performed by an injection-moulding process using a Haake Minijet II (Thermo Scientific, Waltham, MA, USA) with a barrel temperature of 200 °C, mould temperature of 50 °C, injection pressure at 850–1100 bar for 10 s, and holding pressure at 400 bar for 5 s. The synthesis process is shown in Figure 2.

### 2.6. Characterisation and Measurements

#### 2.6.1. Thermogravimetric Analysis (TGA)

TGA experiments were performed using a Q-500 (TA Instruments) under nitrogen atmosphere. A weight of samples of about 5 mg was used to measure the thermal stability of PU/GNP and PU/GO nanocomposites via their heating to 700 °C at a heating rate of 10 °C/min.

#### 2.6.2. Differential Scanning Calorimetric Analysis (DSC)

Q100 (TA Instruments) was used to investigate the thermal properties of PU/GNP nanocomposites using an auto-sampler and the Indium Calibration Standard. The samples weighed about 7–11 mg, and the custom protocol that was applied consisted of a cool/heat/cool/heat sequence with a heating range of −90 °C to 200 °C and a rate of 10 °C/min in an N_2_ atmosphere.

#### 2.6.3. X-ray Diffraction (XRD)

For X-ray diffraction (XRD), an X’Pert X-ray diffractometer was utilized together with a Cu Kα radiation source (λ = 1.542 Å) to first identify the GNP and GO structures in a powder form. The crystallinity plane degree of semicrystalline PU/GNP and PU/GO nanocomposites was attained as well that for injection-moulded samples with a rectangular shape of 1 mm × 0.5 mm.

#### 2.6.4. Small-Angle X-ray Scattering (SAXS)

SAXS experiments were carried out on Beamline I22 at the Diamond Light Source (DLS) facility in Didcot, UK. The beam’s energy was 12.4 keV, equivalent to an X-ray wavelength of 0.1 nm. Samples were suspended on a metal grid using Kapton tape. Acquisition time varied from 1 to 0.01 s depending on the scattering intensity of the sample; the pixel array detector used to collect the SAXS data was the Pilatus P3-2M (from Dectris). There was a 3.47 m fixed distance between samples and the detector, resulting in a momentum transfer vector range of 0.05 (nm^−1^) < q = (4π/λ)sin(θ/2) < 3.0 (nm^−1^), where θ is the scattering angle and λ the wavelength of incident photons. Calibration of the momentum transfer was performed using silver behenate powder. Air was applied as background and was subtracted from all measurements, while the subtraction mask was formed using glassy carbon. Data were reduced using the processing tools in the DawnDiamond software suite. The 2D scattering photon patterns were integrated using an azimuthal integration tool to obtain 1D scattering patterns. As a result of the irregular structures of semi-crystalline materials such as PUs, SAXS can provide a scattering signal with different homogeneities at a typical length scale (d-spacing). This is calculated using Bragg’s law: d_m_ = 2π/q*, where q* is the maximum of the scattering peak. Based on Bragg’s law, the degree of phase separation of multi-block copolymers can be predicted from a one-dimensional scattering pattern for most quantitative analysis; the phase structure can be obtained from the position of the peak of a q scale. This q vector can yield the correlation length between two different phases.

#### 2.6.5. Transmission Electron Microscopy (TEM)

The morphological features of PU/GNP and PU/GO were investigated via a transmission electron microscope (Philips CM200, 200 kV, Tokyo, Japan). For TEM measurement, the PU/GNP and PU/GO samples were prepared via cutting sample into slices approximately 50–60 nm in thickness, which were cut from the core of the samples using a diamond blade knife and an ultra-microtome machine (Leica EM UC6). Samples were deposited on square mesh copper TEM grids (300 Mesh)—purchased from Agar Scientific (UK)—in the distilled water solution.

#### 2.6.6. Scanning Electron Microscopy (SEM)

The morphological features of PU/GNP and PU/GO nanocomposites were detected using field-emission SEM (FESEM-CXL30). The cryogenically fractured surface of the nanocomposites was coated with a thin film of gold in order to make it conductive with respect to the incident electron beam, thus producing a better SEM image. This process was conducted using a rotary-pumped coater (Q150R Plus from Quorum (Lewes, UK)).

#### 2.6.7. Mechanical Test

The tensile properties of the pure PU and PU/GNP and PU/GO nanocomposites were tested using a universal testing machine (type an Instron 1122). All samples were tested under specific conditions (humidity ~50%, with a temperature ~25 °C) within a controlled room. The dumbbell-shaped test specimens were made by injection moulding machine according to the ISO 527-2 1BA standard. The samples were kept overnight before testing; the tensile test was performed with crosshead speed of 10 mm/min for at least 5–7 samples.

## 3. Results and Discussion

### 3.1. Characterisation of Structure and Morphology of GNP/GO

#### 3.1.1. SEM and TEM Analysis

The TEM and SEM techniques were used to investigate the structure and morphology of the GO and GNP nanofillers. Figure 3 shows the SEM and TEM images for both GO and GNP. The SEM images of GO show aggregation and folded regions (indicated by red arrows in Figure 3a) due to oxidation and restacking processes, leading to the formation of a crumpled and fluffy structure [41]. In contrast to GO, both the SEM and TEM images of the GNP reveal folded, rolled up, and overlapping GNP flakes. It is well known that sonication energy can trigger the folding of GO/GNP flakes, which restack after sonication [11,12]. This outcome is attributed to the effect of Van der Waals forces on graphene sheets, leading to restacking and disordered, aggregated regions in the graphene derivatives [42]. Figure 3d illustrates the TEM images of GO at a nano-size thickness and a small scale, which also indicates the folded regions besides the wrinkled regions owing to the effect of the chemical treatment.

#### 3.1.2. XRD Results

Figure 4 presents the XRD diffraction pattern of GNP and GO compared with pure graphite. It is well known that the XRD test is usually used to investigate the crystal structures of different materials and determine their interlayer spacing (d) using Bragg’s equation, where distance (d) relates to the beam-diffraction angles of an incident X-ray line. The XRD diffraction pattern of the as-received GNP exhibits two peaks located at 26.52° and 54.62°, corresponding to Bragg’s reflection planes 002 and 004 and interlayer d-spacing of 3.35 Å and 1.68 Å, respectively [18,43]. As expected from previous studies [43,44], both peaks of the GNPs indexed to pure graphite, which indicates the ability of the as-received GNPs to maintain the crystalline structure of graphite [44,45]. Yet, after oxidation, the highest diffraction peak of graphite shifted to 10.82°, which indicates the effect of the presence of oxygen within the interlayer spaces of graphite [46]. The effect of oxygen functional groups such as carbonyl, hydroxyl, carboxyl, and epoxy groups in GO and some structural defects cause the increase in d-spacing of the highest peak of graphite from 3.35 Å to 8.18 Å, as reported in previous studies [47].

### 3.2. TGA Results

All the gravimetric measurements of pure PU and its nanocomposites with GNP and GO are shown in Figure 5. Figure 5a,c display the normal TGA curves of the PU, PU/GNP, and PU/GO nanocomposites with various weight percentages. The weight loss is plotted with a temperature range from 0 °C to 700 °C. It can be seen that thermal degradation of the polymers can be divided into three steps: the complete degradation of the main chains, the rupture of the side chains with volatile products, and the formation of char residues [48,49]. It was observed that the onset temperature (T_onset_) of pure PU started increasing after the addition of GNPs and GO, and thus both the PU/GNP and PU/GO samples showed a similar trend of thermal stability enhancement, even at a low nanofiller loading. Such a difference in the improvement of the thermal stabilities of the PU nanocomposites via the addition of GNPs and GO is related to the high thermal stability of GNPs compared to GO [50,51,52]. The TGA results can provide evidence of the quality of the nanofillers that are linked with the polymer. Overall, all the PU filled with GNPs and GO exhibited higher thermal stability performance than that of pure PU. The maximum rate of degradation temperature (T_max_) of PU and its nanocomposites as obtained from a DTG thermogram are summarised in Table 1. The DTG curves of PU and its nanocomposites (see Figure 5c,d) consist of a two-stage mechanism of thermal degradation. Firstly, the dissociation of the urethane linkage of PU occurs, converting the urethane groups to isocyanate and alcohol. Secondly, the degradation is associated with the mass loss of the soft segments [18,53].

The findings suggest that PU is thermally stable around 300 °C, while hard segments start to degrade after temperatures in the range of 330 °C–340 °C. The soft segments start to degrade afterwards, with a maximum temperature of 370 °C. The addition of GNP and GO gradually increases the onset temperature by 8 °C and the T_max_ of the DTG results by 10 °C with minimum GNP and GO incorporation. The improved thermal stability results from the maximising of the physico-chemical properties due to the incorporation of nanofillers, since the barrier effect of PU filled with GNP and GO might be more effective in preventing the emission of degraded products, thus yielding high barrier performance [54,55,56,57]. As a result, this tortuous effect prevented the escape of all of the gases generated during the heating process and released from the PU/GNP and PU/GO samples [22,58,59].

#### 3.2.1. DSC Results

DSC was carried out in order to further study the thermal behaviour, including the melting temperature (T_m_), crystallisation temperature (T_c_), melting enthalpies (ΔH_m_), and crystallisation enthalpies (ΔH_C_) of PU and its nanocomposites. Thermal transition variations were investigated by the incorporation of GNP and GO using DSC thermograms, as summarised in Table 2. DSC can provide evidence of the crystal structure of hard segments through the change in the thermal behaviour of the PU/GNP and PU/GO nanocomposites in comparison with neat PU. The DSC scans were conducted with PU nanocomposites containing GNP/GO at different weight loadings (0.25, 0.5, and 0.75 wt.%). The results show that multi-endotherms of the melting temperatures can be detected within semi-crystalline PU. For example, the first heating scan reveals that the endothermic peaks relevant to the melting temperature (T_m_) occurred at around 171 °C for pure PU and then increased to 174 °C and 183 °C for PU containing GNP and GO at 0.75 wt.%. However, the first heating scan illustrates several variations between the first and second heating scan due to the thermal history [60,61]. The second heating scan appears to be more precise compared to the first scan, as the manufacturing parameters varied during the production process owing to the absence of the thermal history [14,48,49]. Thus, an improvement in the PU nanocomposites can be seen in the presence of GNP and GO [7]. The enthalpy of fusion of the second heating scan was observed to increase with the GNP loading (ΔH_m_ rose from 25.44 J/g to 26.4, 27.1, and 29 for loadings of 0.25, 0.5, and 0.75%, respectively). This increase in such an endothermic transition could be attributed to the restricted mobility of the PU chains in the presence of nanofillers [26,62]. It is hard to obtain the glass transition temperature (T_g_) of hard segments due to their mixing with the melted soft segments despite the low ratio of the last one. The presence of multiple peaks in the DSC analysis could refer to several melting–crystallisation processes of the originally formed long-ordered/short-ordered structures of the predominant phase (hard phase) [62,63]. The maximum enhancement in terms of the T_m_ was seen for higher loadings (0.75 wt.%) of both GNP and GO. Furthermore, the crystallisation process was accelerated for the hard segment of PU and was significantly affected by the uniform dispersal of the GNP and GO nanofillers. Since the nanofillers could act as a heterogeneous nucleation agent and thus triggered the better crystallisation of the hard segments of the PU matrix [58], it was found that the T_c_ of the hard domains was greatest and enhanced to almost 120.4 °C and 177.61 °C at loadings of 0.75 wt.% of GNP and GO within the PU matrix, respectively, from approximately 118 °C of pure PU. These findings can be attributed to the larger size of GO and the oxygen-rich functionalities in its structure; therefore, it can act as a positive heterogeneous agent to a greater extent than GNP and cause the HS to improve its intrinsic crystallization within the PU matrix [14]. Meanwhile, the crystallisation enthalpy of the nanocomposites with GO is lower than those with GNP due to the heterogeneous nucleation effect as interpreted in the previous sentence. Similar results regarding DSC were found in recent studies [52,58].

The degree of crystallinity of neat PU and its nanocomposites can be calculated using the following equation:(1)$$X_{C}\%=\frac{\Delta H_{m}}{\Delta H_{m0}\left(1-\emptyset \right)}\times 100$$
where Δ*H_m_* = the heat of fusion of the neat TPU-70 HS and its nanocomposites; Δ*H_m_*_0_ = the heat of fusion for the 100% crystalline TPU, which is recorded as 172.2 J/g and is in accordance with a previous study [64]; and finally, *∅* = the nanoparticle fraction in the TPU structure [64].

Thus, through this equation, the degree of crystallinity of pure PU is about 14.7%, which increased to reach 25.7% and 23.6% for the PU nanocomposites incorporated with 0.75 wt.% of GNP and GO, respectively. Even though the PU/GO nanocomposites recorded a higher Tc in comparison with GNP, its degree of crystallinity was slightly lower than that of the PU/GNP nanocomposites at the same 0.75 wt.% loading. This can be assigned to the higher surface area of GO and the presence of oxygen functionality in its structure, leading to an increase in ΔHm and then a decrease in Xc according to the above equation [65,66]

#### 3.2.2. Wide-Angle X-ray Diffraction (WAXS) Results

The wide-angle X-ray diffraction detected the long-range order and regularity of the PU chains, particularly the hard-segment chains, as illustrated in Figure 6. The X-ray diffractograms of the PU structure display two main diffraction peaks at 2θ values of 11° and 2θ values of 20°. The first corresponds to the interlayer spacing of the hard segments of 0.8 nm and 0.43 nm, based on Bragg’s equation [61,67]. The main diffraction peaks of PU represent the long order or even the crystallinity of hard segments. With the addition of a small amount of GNPs or GO, the disappearance of the diffraction peaks of GNP and GO indicate the better exfoliation of the GNP and GO within the PU matrix due to proper dispersion [68,69]. The decrease in the intensity among the GO nanofillers is due to the oxidation process during manufacturing [12,70]. The soft segments in this study did not exhibit any diffraction peaks due to the difficulty of crystallising the polyether polyol. The diffraction of the PU/GNP nanocomposites is clearly observed when adding GNP at various loadings. On the contrary, the PU/GO nanocomposites showed relatively wide diffraction peaks, which indicate the disruption of the long-range order of hard segments and thus a reduction in the crystallinity of PU [71]. In general, the intensity of the diffraction peaks was seen to increase with an increase in the nanofiller (GNP and GO) loading. However, the diffraction peaks vanished in the case of GO incorporation, which is attributed to the presence of functional groups on the surface of GO that might have reacted with the PU chains (hard segments). Similar results to those presented in this work can also be found in previous reports on similar nanofillers [72,73].

### 3.3. SAXS Results

The SAXS pattern of pure PU and its nanocomposites with GNP and GO at different concentrations (0.25 wt.%, 0.5 wt.%, and 0.75 wt.%) is shown in Figure 7. The maximum of the scattering peak (q*) of the microphase-separated pattern is observed in the scattering curve (*I(q)* vs. *q*)) [63,74]. SAXS showed a maximum scattering for all the PU samples at the same momentum transfer, *q*, value, as previously mentioned in a study by Saiani et al. [75]. From SAXS, the phase structure of PU was estimated to be within the length range of 1–100 nm based on the contrast between the electron densities of hard segments and soft segments. Thus, a heterogeneous microstructure of PU was produced, since the scattering intensity was directly proportional to the electron variations (Q_inv_) (microphase separation index) that represented the two-phase system. This could be an indicator of how hard segment interaction/association works to produce the domain size and thus the micro-phase separation of PU [74,75,76].

In the present study, a characteristic scattering peak is observed for all the PU/GNP samples, implying the presence of a microphase-separated structure. The broadening of the peaks toward higher *q* values can prove the assumption of a reduction in interdomain spacing and thus the conclusion that a better phase separation of PU has been induced by the GNPs’ incorporation [63]. This assumption has been proposed previously by Teheran et al. [77]. On the other hand, a similar trend was also observed in the PU/GO nanocomposites with a partial disappearance of the scattering peak, implying that hard segments tend to be packed in a uniform pattern promoted by strong interactions with the uniformly dispersed GO [12,63,77]. In Figure 7a, the characteristic peak for pure PU is found at *q* = 0.45 nm^−1^ and that for PU/GNP with loadings of 0.25, 0.50 and 0.75 wt.% is found at *q* = 1.1 and 0.4 nm^−1^, which corresponds to the d-spacing of pure PU and PU/GO of 13.95, 6.28, 6.28, and 15.7 nm. This current finding is in agreement with the previous XRD results. Conversely, in Figure 7b, a broad peak associated with the characteristic peak of PU/GO appears, showing no peaks for all concentrations. The disappearance of the peak indicates that there is a clear level of dispersion and interaction of GO particles in the polyurethane matrices during high-shear mixing [78]. This phenomenon is based on functional groups on the edge of GO that cause physical and/or chemical interactions between –OH groups in the nanoparticles and carbonyl groups in the polyurethane structures [71,79]. This contributed to the phase separation dropping between HS and SS [78]. Overall, the PU/GNP nanocomposites show a slight shift in the q_max_ towards a higher *q* value at low loadings, demonstrating a less interfacial interdomain region.

### 3.4. SEM Results

It is important to investigate the influence of the GNP and GO nanofillers’ addition on the structural pattern of the PU matrix. The influence of nanofillers depends on how effectively they are dispersed in the PU matrix. A poor dispersion or distribution of GNP and GO can have a significant impact on the PU’s morphology, leading to an inconsistent behaviour. Figure 8 shows a comparison of the SEM images of the cryo-fractured surfaces of the PU/GNP and PU/GO nanocomposites versus pure PU. Figure 8a presents a quite homogeneous and smooth surface of the pure PU. The incorporation of GNP changes the morphology and grants the nanocomposites a more flake–like morphology. The fractured surface of the PU/GNP nanocomposites with increasing proportions of GNP is seen to have some aggregated GNPs and apparent irregularities compared to the surface of pure PU, suggesting the poor dispersibility of the individual GNPs (indicated by the red arrows and ellipses) as illustrated in Figure 8b–d [80]. As a result, the SEM morphology can provide a demonstration of the dispersion quality of the GNP within the PU [3,35]. In the other samples of PU/GO, the GO platelets show better attachment to the PU chains, particularly the hard segments, owing to the interfacial adhesion between the GO surfaces and PU chains, and thus a good enhancement of the dispersion of GO in the PU nanocomposites [22,58,73].

### 3.5. TEM Results

Figure 9 shows the TEM images, demonstrating the nanoscale dispersion of nanofillers within the PU nanocomposites. The dark areas are regions of GNP; these features may be attributed to the linkage of GNP with the PU matrix. The microtome-cut samples at 0.25%, 0.5 %, and 0.75% GNP show well-dispersed GNPs in the PU matrix at a low loading (0.25%). However, the individual GNP tend to aggregate and increase in thickness with increasing loadings. As a result, this is referred to as a strong interfacial interaction between the GNPs and PU due to a proper dispersion technique with a small amount of reinforcement [81,82,83]. In the case of the PU/GO nanocomposites, the TEM images exhibit a uniform dispersion and distribution of GO flakes at a minimum concentration. On the contrary, this trend is seen to change as the amount of GO increases within the PU matrix, which is clear evidence for the tendency of GO towards restacking, despite the existence of strong interactions with the PU chains [84,85,86]. In general, the TEM results show a good agreement with the SEM results regarding the dispersion of GNP and GO within a PU matrix.

### 3.6. Mechanical Performance Results

A comparative study of the mechanical properties of PU nanocomposites with GNPs and GO was conducted. The tensile behaviour of PU when GNP and GO nanofillers were incorporated using in situ polymerisation displayed a good improvement compared to pure PU. These enhancements in tensile properties such as the tensile modulus and yield strength were determined through a stress–strain analysis, as depicted in Figure 10. Based on the overall enhancement of the tensile performance of the PU/GNP and PU/GO nanocomposites, several factors affect the efficiency/quality of tensile improvement, including the GNPs’ and GO’s dispersion, concentration, and bonding ratio within the PU polymer [6,87,88,89]. To some extent, Young’s modulus has been increased from 214 MPa for PU to maximum values of 320 MPa, 390 MPa, and 490 MPa for GNP loadings of 0.25, 0.5, and 0.75 wt.%, respectively. The increments in the percentage of the maximum modulus were recorded as 50%, 81%, and 127%, respectively. Similarly, the PU/GO nanocomposites have the greatest modulus at percentages of 0.25 and 0.5 wt.% compared to the PU/GNP nanocomposites. The possible reason for this substantial improvement could be related to the superior dispersion and interfacial adhesion between the functional groups on the GO surfaces with a high stiffness and the hard segments of PU, which are observed to increase to 136% and 152% for 0.25 and 0.5 wt.%, respectively [58]. However, this increase is seen to reduce at a 0.75% loading of GO. This is because of the high aspect ratio of GO leading to the clustering of GO at higher concentrations and thus increasing the number of stress concentration sites, which reduce the quality of the resultant nanocomposites through a reduction in stress transfer efficiency [2,90,91].

The same results are observed regarding the tensile strength at break: all the tensile values have been increased as a function of the nanofillers’ addition from 16.3 MPa for pure PU to 18 MPa, 23 MPa, and 25 MPa with GNP loadings of 0.25, 0.5, and 0.75 wt.%, respectively. The effective load transfer is attributed to such an improvement in tensile strength with the assumption of an increase in the microphase separation of PU [6,33,90]. A similar trend is observed for the PU/GO nanocomposites, whose increases are 111 MPa, 123 MPa, and 28 MPa for GO loadings of 0.25, 0.5, and 0.75 wt.%, respectively, compared to that of pure PU. A noticeable improvement in the tensile strength of the PU/GO samples was observed compared to that of PU/GNP; this is likely due to the efficient interfacial interaction, as previously mentioned, along with the modulus’ improvement [92].

However, a decrease is observed at a 0.75% loading, suggesting aggregation at a high GO content [89]. The restrictive effect of the GNP/GO nanofillers on the PU chains’ mobility results in a decrement in elongation at break for their nanocomposites. Reduction ratios of 27% and 72% were seen for the maximum loadings of GNP and GO, respectively, compared to pure PU. However, an increase at 0.25 of GO with respect to PU in the elongation at break point was also observed, suggesting a form of soft transition or even a lubricating effect between the GO and PU chains, depending on the dispersibility of GO within PU [3,22].

## 4. Conclusions

This study has successfully synthesised novel PU nanocomposites with two different graphene-based nanofillers (GNP and GO) with a chain extender (1,5 Pentanediol). The authors investigated the effect of the addition of different nano-scale fillers at three different loadings (0.25, 0.5, and 0.75 wt.%) on the mechanical, thermal, and morphological properties of the PU nanocomposites formed via in situ polymerisation. The thermal transitions obtained by DSC showed that the melting temperature (T_m_) and crystallisation temperature (T_c_) have been enhanced by the addition of GNP at any level. The enthalpies of fusion and crystallisation (Δ*H_m_* and Δ*H_C_*) were also improved, meaning that the nanofillers worked as nucleation agents. This result was obtained because the improved dispersion and interaction of GNP can lead to microphase separation, and thus the better crystallisation of hard segments. In the case of the addition of GO, a similar trend was seen, with an enhanced melting point of the resultant PU nanocomposites due to strong interfacial adhesion between the GO and PU chains.

The WAXS results showed an improvement in the crystalline peak of pure PU with the addition of GNP, while the presence of GO caused the diffraction peak to vanish. This is due to the functional group on the surface of GO leading to interfacial interaction with the PU matrix.

The SAXS results indicate that the presence of GNP and GO particularly stimulated an increase in the microphase separation of the PU matrix. SEM and TEM morphologies were used to support the dispersion quality obtained using different reinforcement nanofillers. The micrographs showed that both GNP and GO dispersed effectively within PU at low levels of concentration and some clusters of aggregation were observed at high concentrations. The mechanical tests showed a dramatic increase in the tensile properties as a result of incorporating GNP and GO nanofillers in comparison with the PU polymer. Thus, the improved modulus of the resultant PU nanocomposites was seen to have a greater value at 0.5 wt.% GNP and GO loadings. However, the tensile strength and elongation at break decreased after this percentage due to the suppression of strain hardening.

## Figures and Tables

**Figure 1 polymers-14-04224-f001:**
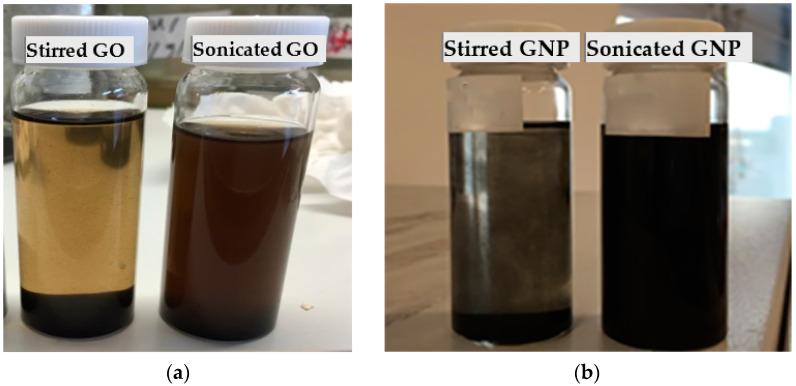
Dispersibility behaviour of GNP and GO in DMAC solvent (0.5 mg/mL) after ultrasonication and magnetic stirring-processes and being left for 24 h: (**a**) GO (**b**) GNP.

**Figure 2 polymers-14-04224-f002:**
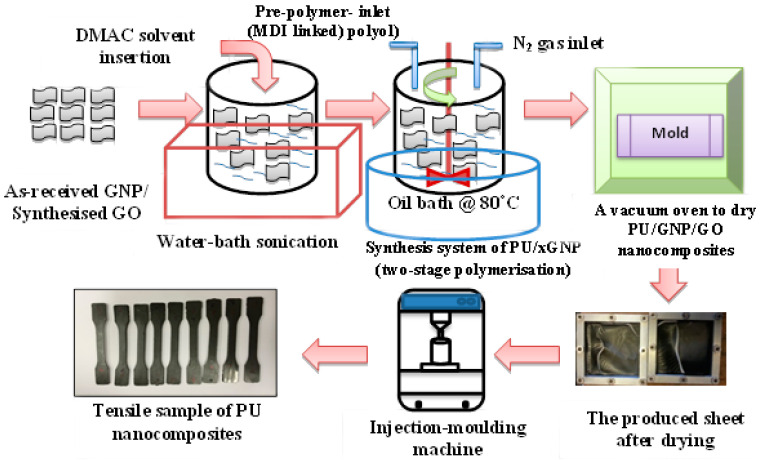
Schematic representation of fabrication process of PU/GNP and PU/GO nanocomposites using a sonication bath and in situ polymerisation system with an oven and nitrogen inlet to maintain the inert atmosphere environment inside the reaction vessel.

**Figure 3 polymers-14-04224-f003:**
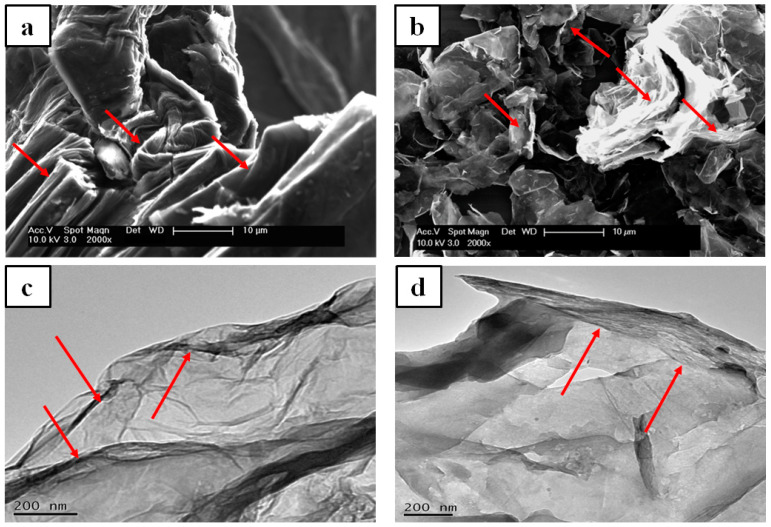
SEM and TEM images: (**a**,**c**)—GO; (**b**,**d**)—GNP. The red arrows indicate the folding and wrinkling areas and aggregation regions of GO.

**Figure 4 polymers-14-04224-f004:**
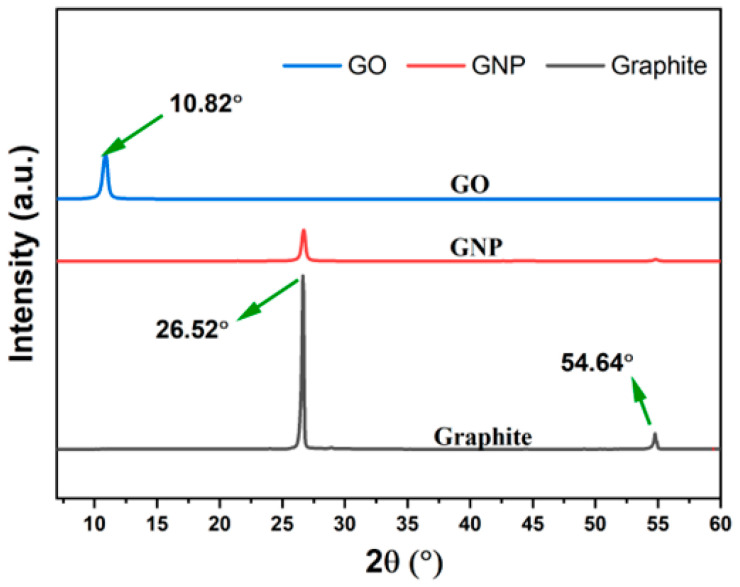
XRD pattern of GNPs and GO in comparison with pure graphite.

**Figure 5 polymers-14-04224-f005:**
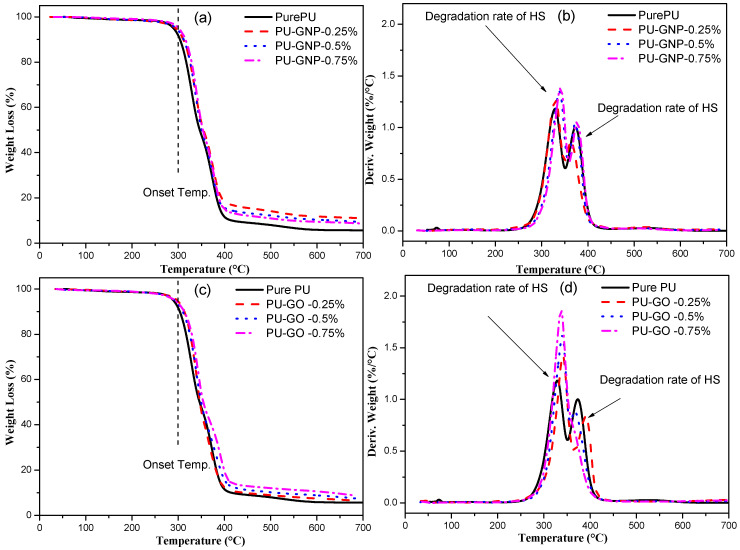
TGA curves (10 °C/min) of PU/GNP nanocomposites and PU/GO nanocomposites compared to pure PU: (**a**) TGA curves; (**b**) DTG curves.with GNP nanofiller, (**c**) TGA curves; (**d**) DTG curves, with GO nanofiller.

**Figure 6 polymers-14-04224-f006:**
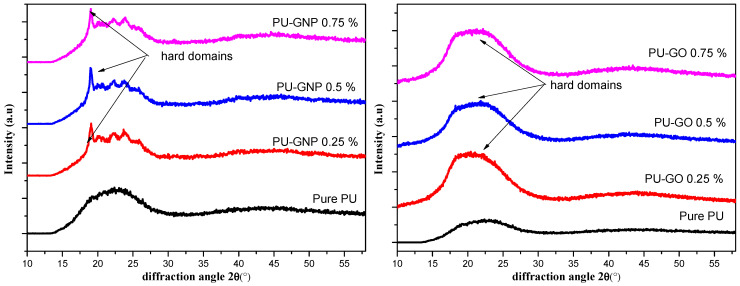
Semi-log plot of Y-axis of wide-angle X-ray-scattering intensity for PU/GNP/GO nanocomposites.

**Figure 7 polymers-14-04224-f007:**
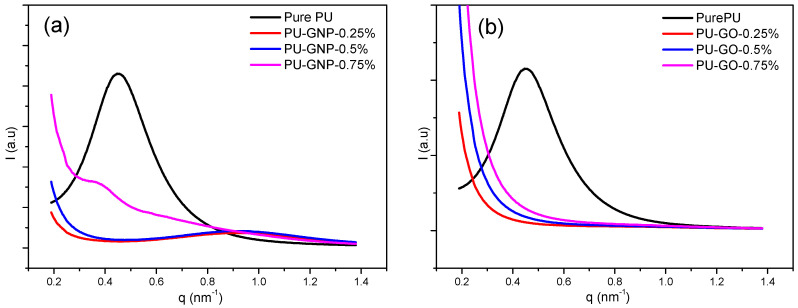
Semi-log plot of scattered intensity from SAXS experiments for pure PU and PU/GNP and PU/GO nanocomposites, (**a**) SAXS curves of GNP nanofiller; (**b**) SAXS curves of GO nanofiller.

**Figure 8 polymers-14-04224-f008:**
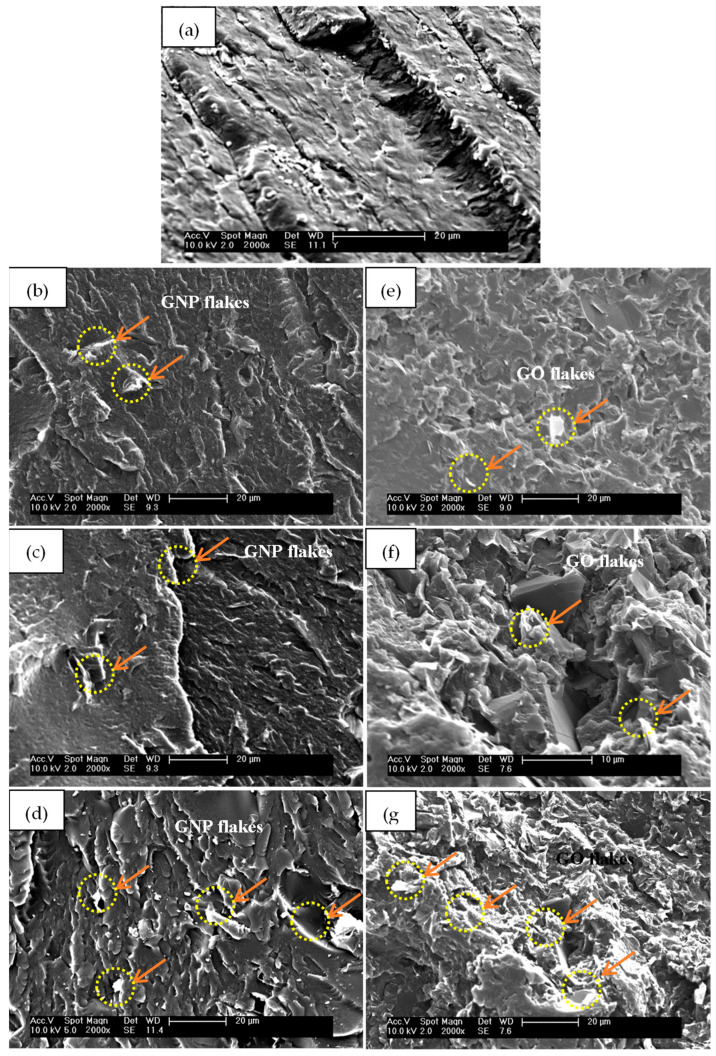
SEM images of cryogenic-fracture surfaces of (**a**) pure PU, (**b**–**d**) PU/GNP, and (**e**–**g**) PU/GO nanocomposites with loadings of GNP and GO of 0.25 wt.%, 0.5 wt.%, and 0.75 wt.%, respectively.

**Figure 9 polymers-14-04224-f009:**
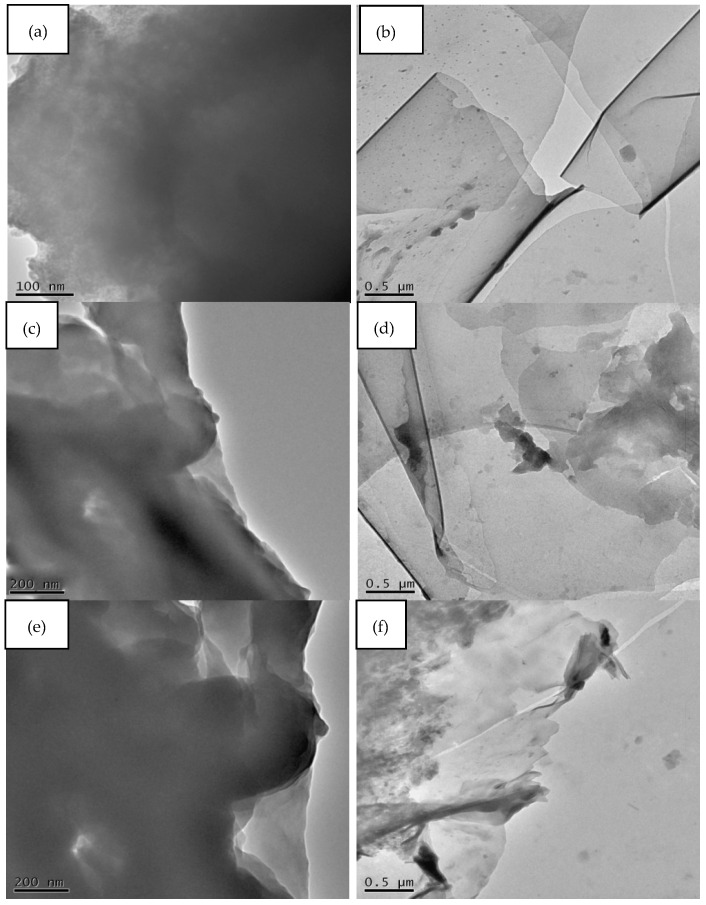
TEM images: (**a**,**c**,**e**) PU/GNP; (**b**,**d**,**f**) PU/GO nanocomposites with loadings of 0.25 wt.%, 0.5 wt.%, and 0.75 wt.%, respectively, with a magnification scale of 200 µm and 0.5 µm.

**Figure 10 polymers-14-04224-f010:**
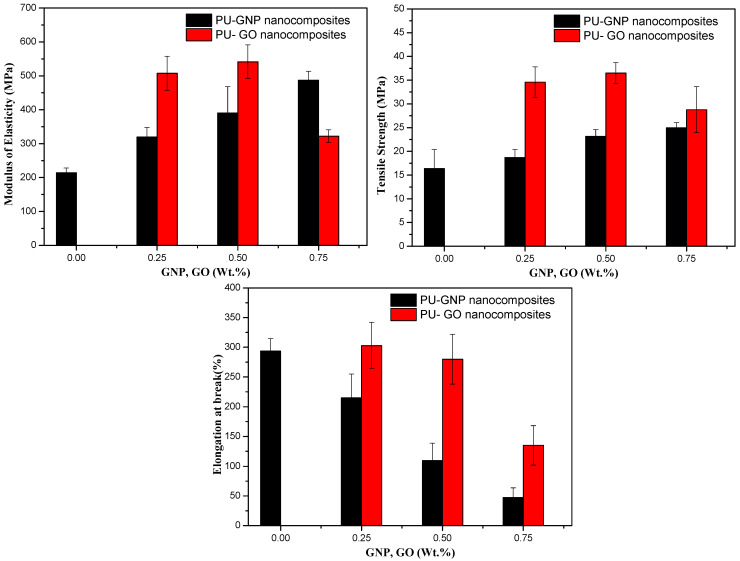
Comparative mechanical properties of the PU/GNP and PU/GO nanocomposites.

**Table 1 polymers-14-04224-t001:** TGA data for GNP- and GO-filled PU.

	TGA				DTG							
	T _Onset_ ± SD		Residual Mass (%)@600 °C ± SD		1st Step (HS Degradation)				2nd Step (SS Degradation)			
	GNP	GO	GNP	GO	T_max_ (°C) ± SD		Dev. Mass (%/°C) ± SD		T_max_ (°C) ± SD		Dev. Mass (%/°C) ± SD	
					GNP	GO	GNP	GO	GNP	GO	GNP	GO
**Neat PU**	297.76 ± 1.10	297.76 ± 1.10	8.50 ± 2.88	8.50 ± 2.88	333.70 ± 1.34	333.70 ± 1.34	1.16 ± 0.13	1.16 ± 0.10	369.00 ± 2.45	369.00 ± 2.45	0.83 ± 0.11	0.83 ± 0.11
**PU + 0.25 (wt.%)**	308.7 ± 3.73	307.67 ± 3.80	10.11 ± 0.03	6.76 ± 1.30	334.80 ± 2.51	342.14 ± 1.80	1.24 ± 0.03	1.50 ± 0.05	368.20 ± 4.01	380.80 ± 1.40	0.92 ± 0.09	0.95 ± 0.02
**PU + 0.5 (wt.%)**	309.20 ± 4.34	308.90 ± 1.50	10.5 ± 1.13	7.79 ± 0.87	338.67 ± 1.40	343.57 ± 2.40	1.38 ± 0.11	1.8.80 ± 0.10	372.86 ± 3.30	382.88 ± 2.70	0.99 ± 0.03	1.1 ± 0.65
**PU + 0.75 (wt.%)**	314.56 ± 0.55	315.46 ± 2.30	9.62 ± 1.62	10.50 ± 0.27	341.47 ± 2.02	343.6 ± 2.70	1.41 ± 0.12	1.34 ± 2.8	375.80 ± 2.02	392.45 ± 2.49	1.02 ± 0.05	1.30 ± 0.77

**Table 2 polymers-14-04224-t002:** Thermal transitions of PU/GNP and PU/GO nanocomposites compared to pure PU.

	First Heating				First Cooling				Second Heating			
	GNP		GO		GNP		GO		GNP		GO	
	T_m_ ± SD	ΔH_m_ ± SD	T_m_ ± SD	ΔH_m_ ± SD	T_c_ ± SD	ΔH_c_ ± SD	T_c_ ± SD	ΔH_c_ ± SD	T_m_ ± SD	ΔH_m_ ± SD	T_m_ ± SD	ΔH_m_ ± SD
**Pure PU (65% HS)**	171.60 ± 1.85	25.44 ± 1.39	-	-	118.31 ± 1.54	27.49 ± 0.09	-	-	174.29 ± 1.62	26.11 ± 0.07	-	-
**PU + 0.25 (wt.%)**	172.3 ± 0.60	26.39 ± 2.2	183.78 ± 2.77	27.59 ± 2.02	121.01 ± 1.40	28.195 ± 1.60	175.53 ± 2.60	23.07 ± 1.00	177.43 ± 0.60	25.06 ± 2. 70	176.71 ± 3.10	-
**PU + 0. 50 (wt.%)**	172.60 ± 1.65	27.13 ± 1.34	183.73 ± 0.00	29.67 ± 1.33	120.20 ± 1.30	30.89 ± 0.67	176.62 ±1.90	25.61 ± 1.50	175.62 ± 0.67	26.13 ± 1.40	178.94 ± 2.00	-
**PU + 0.75 (wt.%)**	174.34 ± 1.00	26.80 ± 1.20	183.21 ± 3.05	29.13 ± 0.90	120.70 ± 2.70	33.10 ± 1.30	177.61 ± 0.00	29.87 ± 0.60	176.90 ± 0.90	26.78 ± 0.60	179.55 ± 2.60	-

## Data Availability

All research data supporting this work are directly available within this publication.

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
