# Peer review of "Thermal, Mechanical, and Morphological Characterisations of Graphene Nanoplatelet/Graphene Oxide/High-Hard-Segment Polyurethane Nanocomposite: A Comparative Study"

_polymers, 2022, doi:10.3390/polym14194224_

Round 1
Reviewer 1 Report
Albozahid et al. in their manuscript entitled “Thermal, Mechanical, and Morphological Characterizations of Graphene Nanoplatelet/Graphene Oxide/High Hard-Segment Polyurethane Nanocomposite: A Comparative Study”, investigated the effect of the addition of graphene nanoplatelets (GNPs) and graphene oxide (GO) to the polyurethane on its thermal, morphological, and mechanical properties using DSC, TGA, XRD, TEM, SEM, and mechanical tests. However, some deficiencies listed below are observed:
1. The manuscript contains many typos/spelling mistakes. They should be corrected.
2. The text in Figure 1 is unclear. The authors should rewrite it. However, all images were taken from the author’s previous work without any modification [https://doi.org/10.1007/s10965-021-02660-5].
3. “Dispersion Techniques” step should be rearranged before the “Preparation of PU/GNP, PU/GO Nanocomposites” step.
4. The calculation methods of the fillers (GNP, GO) ratio in the prepared nanocomposites are absent. The authors should explain how they determined the wt. percentage of the added fillers.
5. The characterization and conditions of the mechanical tests are absent in the manuscript. The authors should add them.
6. The authors wrote “Figure 3b illustrates the TEM images of GO in nano-size thickness and small scale”, Is this figure related to TEM or SEM?
7. The aim of the XRD measurements and presented results (section 3.1.2) is not clear. The presented information is already known. The authors should explain what the main purpose of this test is.
8. The authors wrote “gravimetric measurements of pure PU and its nanocomposites with GNP and GO are shown in Figure 3”. The numbering of the figures is wrong, the authors should correct the numbering for all figures. Moreover, the figure presented the gravimetric measurements contains some mistakes that should be corrected.
9. Depending on DSC results, the discussion about the relation between Tm and crystals size is absent. What do the authors mean by writing “The enhancement in Tm”? Does this mean an increase in Tm and why? However, the discussion related to the obtained DSC results should significantly improve. The authors only presented the values without a deep discussion. Moreover, both GO and GNP can act as heterogeneous nucleation centers. The authors should discuss the effect of filler addition on the crystallinity and crystals size.
10. Can the authors determine the crystallinity of the obtained nanocomposites from the obtained WAXS results?
11. Comparing to lots of articles related to polyurethane/graphene nanocomposites, what is the novelty of this work? The authors should clarify what is the new in their manuscript.
12. The references should be rewritten according to MDPI and ACS style.
In general, the article is good in technical terms, but, unfortunately, is considered somehow weak in scientific terms. The authors should improve the discussion for all results obtained.
Author Response
REVIEWER 1
- The manuscript contains many typos/spelling mistakes. They should be corrected.
Answer: Thank you for your advice. All typos/spelling mistakes have been corrected (please see tracked corrections)
- The text in Figure 1 is unclear. The authors should rewrite it. However, all images were taken from the author’s previous work without any modification [https://doi.org/10.1007/s10965-021-02660-5].
Answer: Thank you for your suggestion. It is not taken from our previous article as we have modified it. However, the text in Figure has been clarified as recommended.
- “Dispersion Techniques” step should be rearranged before the “Preparation of PU/GNP, PU/GO Nanocomposites” step.
Answer: Thank you for suggestion. It has been modified according to your suggestion (please see section 2.4)
- The calculation methods of the fillers (GNP, GO) ratio in the prepared nanocomposites are absent. The authors should explain how they determined the wt. percentage of the added fillers.
Answer: Thank you for your suggestion. It has been added to the article in the experimental part (please see section 2.5. Preparation of PU/GNP, PU/GO Nanocomposites)
- The characterization and conditions of the mechanical tests are absent in the manuscript. The authors should add them.
Answer: Thank you for your suggestion. It has been added to the article in the experimental part (please see section 2.6.7.)
- The authors wrote “Figure 3b illustrates the TEM images of GO in nano-size thickness and small scale”, Is this figure related to TEM or SEM?
Answer: Thank you for your notice. It has been corrected according to your suggestion (please see page 6 in results and discussion)
- The aim of the XRD measurements and presented results (section 3.1.2) is not clear. The presented information is already known. The authors should explain what the main purpose of this test is.
Answer: Thank you for notice. It has been explained and added within the XRD result,“As known that the XRD test usually used to investigate the crystal structures of different materials and determine their interlayer spacing (d) using Bragg’s law equation, where this distance (d) relates to the beam-diffraction angles of an incident x-ray line.” (Please see page 6 in results and discussion)
- The authors wrote “gravimetric measurements of pure PU and its nanocomposites with GNP and GO are shown in Figure 3”. The numbering of the figures is wrong, the authors should correct the numbering for all figures. Moreover, the figure presented the gravimetric measurements contains some mistakes that should be corrected.
Answer: Thank you for your notice. It has been corrected.
- Depending on DSC results, the discussion about the relation between Tmand crystals size is absent. What do the authors mean by writing “The enhancement in Tm”? Does this mean an increase in Tm and why? However, the discussion related to the obtained DSC results should significantly improve. The authors only presented the values without a deep discussion. Moreover, both GO and GNP can act as heterogeneous nucleation centers. The authors should discuss the effect of filler addition on the crystallinity and crystals size.
Answer: Thank you for your suggestion. The enhancement of Tm of PU polymer can be defined as an increase in Tm. In fact, the most recent article relevant to our study calculated the value of crystallinity from the DSC scan. So, the influence of fillers on hard segment crystallinity was determined by DSC. However, there are many parameters that can be affected the crystal size of PU chains including; the type of PU itself (the amount of hard segments/soft segments as well as the type of raw materials for this polymer such as the type of isocyanate and polyol).also, the degree of phase mixing and demixing could influence the degree of crystallinity of PU. Therefore, for precise investigation of crystallinity, DSC data have been used to calculate and find the value of crystallinity from the thermal enthalpies from DSC run and supported by previous studies. (Please see the DSC discussion in page 10)
- Can the authors determine the crystallinity of the obtained nanocomposites from the obtained WAXS results?
Answer: Thank you for your suggestion. It can be calculated by knowing the amount of amorphous and crystalline parts in the PU in the nanocomposites .The degree of crystallinity is finally calculated by: Xc =TS- A /TS where Ts is the total scattered intensity and A is the scattering from the amorphous halo. Moreover, a deconvolution analysis was performed. This was obtained by fitting Lorentzian functions, in proximity of each characteristic reflection. (please see our previous article doi/abs/10.1080/25740881.2021.1991952). Some researchers found that the calculation of the degree of crystallinity from DSC results was more accurate than XRD results, one of them is the study of M. Doumeng et al (2020) https://doi.org/10.1016/j.polymertesting.2020.106878. The authors concluded that “The XRD underestimates the degrees of crystallinity which are lower than the values determined by density at high degrees of crystallinity. On the one hand, it is difficult, for low crystallinity rates, to extract the crystalline part hidden by the halo from the amorphous phase. On the other hand, the quantity of amorphous phase is difficult to estimate on the samples of which the crystalline part is important. The degree of crystallinity is possibly lowered.”
- On the other hand, many researches which used the XRD results to calculate the XC noticed via that the intensity and sharpness of the crystalline peaks are not the exact amount of crystalline fraction. Other previous studies referred to this finding such as Amir Rostami, 2019 with the context “The addition of fCNTs and fGnPs narrows the crystalline peak of the TPU hard domain, which is assumed to indicate an increased in degree of crystallinity”and Hosseini Hosseini-Sianaki et al., 2015 “This narrowing effect which is believed to be an indication of increasing the degree of crystallinity , supports our argument on the thermal analysis results regarding the nucleating effect of MWCNT on crystallization taking place in the hard domain as a part of MWCNT enhancing role on the microphase separation. In this way, MWCNTs assist more fraction of noncrystalline part of hard domains to regulate their structure and change it to crystalline phase”. Thus, the degree of crystallinity can be calculated either by DSC or XRD technique but it is difficult to compare the results of both techniques to each other due to the difference in principal, technicism, sample size and temperature for both techniques.
- Comparing to lots of articles related to polyurethane/graphene nanocomposites, what is the novelty of this work? The authors should clarify what is the new in their manuscript.
Answer: Thank you for your comments. It is clear that there is a lack in previous studies to investigate this kind of PU with a different type of chain extender (1,5 PD) and hard segments size as nanocomposites with these types of nanofillers and such fraction percentage
- The references should be rewritten according to MDPI and ACSstyle.
Answer: Thank you for your suggestion. It has been modified according to your suggestion
Best regard,

Reviewer 2 Report
This manuscript was well organized and written, and I suggest to accept after addressing the following issues:
1. The authors are suggested to cite the following references in the first paragraph of Introduction:
(1) Adv. Funct. Mater., 2020, 30(17): 1908101.
(2) Compos. Part B-Eng., 2019, 167: 356.
(3) Polym. Composite., 2021, 42(4): 1698.
(4) Polymers, 2022, 14(6): 1205.
2. Please describ '2.6. Characterisation and Measurements' in more brief sentences, especially for SAXS.
3. The scale bars in Figure 3a and 3b should be remodified since they were not as clear as those in other figure.
4. The temperatures could not be as accurate as presented in Table 1 and Table 2, and the authors are suggested to keep one decimal place.
5. 'Elongation' in Figure 8 should be revised to 'Elongation at break'.
6. The detailed experimental description about mechanical performance should be added, and please define modulus of elasticity in Figure 8.
Author Response
REVIEWER 2
Comments and Suggestions for Authors
This manuscript was well organized and written, and I suggest to accept after addressing the following issues:
- The authors are suggested to cite the following references in the first paragraph of Introduction:
(1) Adv. Funct. Mater., 2020, 30(17): 1908101.
(2) Compos. Part B-Eng., 2019, 167: 356.
(3) Polym. Composite., 2021, 42(4): 1698.
(4) Polymers, 2022, 14(6): 1205.
Answer: Thank you for your suggestion.
- Please describ '2.6. Characterisation and Measurements' in more brief sentences, especially for SAXS.
Answer: Thank you for your suggestion. It has been modified
- The scale bars in Figure 3a and 3b should be remodified since they were not as clear as those in other figure.
Answer: Thank you for your suggestion. It has been modified
- The temperatures could not be as accurate as presented in Table 1 and Table 2, and the authors are suggested to keep one decimal place.
Answer: Thank you for your suggestion. It has been modified
- 'Elongation' in Figure 8 should be revised to 'Elongation at break'.
Answer: Thank you for your suggestion. It has been modified
- The detailed experimental description about mechanical performance should be added, and please define modulus of elasticity in Figure 8.
Answer: Thank you for your notice. It has been added. Modulus of elasticity can be defined as the resistance of a material to elastic deformation, and represents the ratio of the stress in a body to the corresponding strain
Best regard,

Round 2
Reviewer 1 Report
I would like to suggest accepting in present form.